# Risk Management and Patient Safety in the Artificial Intelligence Era: A Systematic Review

**DOI:** 10.3390/healthcare12050549

**Published:** 2024-02-27

**Authors:** Michela Ferrara, Giuseppe Bertozzi, Nicola Di Fazio, Isabella Aquila, Aldo Di Fazio, Aniello Maiese, Gianpietro Volonnino, Paola Frati, Raffaele La Russa

**Affiliations:** 1Department of Anatomical, Histological, Forensic and Orthopaedic Sciences, Sapienza University of Rome, 00161 Rome, Italy; michela.ferrara@uniroma1.it (M.F.); nicola.difazio@uniroma1.it (N.D.F.); paola.frati@uniroma1.it (P.F.); 2Complex Intercompany Structure of Forensic Medicine, 85100 Potenza, Italy; giuseppe.bertozzi@unifg.it; 3Department of Medical and Surgical Sciences, University Magna Graecia of Catanzaro, 88100 Catanzaro, Italy; isabella.aquila@unicz.it; 4Regional Hospital “San Carlo”, 85100 Potenza, Italy; aldo.difazio@ospedalesancarlo.it; 5Department of Surgical Pathology, Medical, Molecular and Critical Area, Institute of Legal Medicine, University of Pisa, 56126 Pisa, Italy; aniello.maiese@unipi.it; 6Department of Clinical Medicine, Public Health, Life and Environment Science, University of L’Aquila, 67100 L’Aquila, Italy; raffaele.larussa@univaq.it

**Keywords:** artificial intelligence, clinical risk management, patient safety, machine learning

## Abstract

Background: Healthcare systems represent complex organizations within which multiple factors (physical environment, human factor, technological devices, quality of care) interconnect to form a dense network whose imbalance is potentially able to compromise patient safety. In this scenario, the need for hospitals to expand reactive and proactive clinical risk management programs is easily understood, and artificial intelligence fits well in this context. This systematic review aims to investigate the state of the art regarding the impact of AI on clinical risk management processes. To simplify the analysis of the review outcomes and to motivate future standardized comparisons with any subsequent studies, the findings of the present review will be grouped according to the possibility of applying AI in the prevention of the different incident type groups as defined by the ICPS. Materials and Methods: On 3 November 2023, a systematic review of the literature according to the Preferred Reporting Items for Systematic Reviews and Meta-Analyses (PRISMA) guidelines was carried out using the SCOPUS and Medline (via PubMed) databases. A total of 297 articles were identified. After the selection process, 36 articles were included in the present systematic review. Results and Discussion: The studies included in this review allowed for the identification of three main “incident type” domains: clinical process, healthcare-associated infection, and medication. Another relevant application of AI in clinical risk management concerns the topic of incident reporting. Conclusions: This review highlighted that AI can be applied transversely in various clinical contexts to enhance patient safety and facilitate the identification of errors. It appears to be a promising tool to improve clinical risk management, although its use requires human supervision and cannot completely replace human skills. To facilitate the analysis of the present review outcome and to enable comparison with future systematic reviews, it was deemed useful to refer to a pre-existing taxonomy for the identification of adverse events. However, the results of the present study highlighted the usefulness of AI not only for risk prevention in clinical practice, but also in improving the use of an essential risk identification tool, which is incident reporting. For this reason, the taxonomy of the areas of application of AI to clinical risk processes should include an additional class relating to risk identification and analysis tools. For this purpose, it was considered convenient to use ICPS classification.

## 1. Introduction

Healthcare systems represent complex organizations [1] within which multiple interconnected factors (physical environment, human factors, technological devices, quality of care) form a dense network whose imbalance is potentially able to compromise patient safety [2]. The latter, defined as “the absence of preventable harm to a patient and reduction of risk of unnecessary harm associated with healthcare to an acceptable minimum” [3], is a fundamental principle of healthcare and is part of the patient’s rights. According to the World Health Organization (WHO), there is a one in a million chance of aviation accidents, while the possibility of harming a patient in the Clinical process is one in three hundred [3]. As specified by the Global Patient Safety Action plan 2021–2030, patient safety incidents are a growing problem and one of the major causes of death and disability worldwide [4]. In this scenario, the need for hospitals to expand clinical governance programs is easily understood. Clinical governance is defined as the system through which healthcare organizations improve the quality of care and guarantee high standards of care, striving for excellence [5].

One of the main pillars of this healthcare quality system is clinical risk management, which refers to the set of proactive and reactive clinical tools, procedures, and processes used to detect, monitor, reduce, and prevent potential risks and errors in clinical practice to safeguard patient safety [6]. Risk assessment instruments include incident reporting [7], reviews of medical records, safety walk-arounds, administrative data obtained from hospital discharge forms, patient complaints, and information derived from claims litigation. These methods allow for the identification of potential or actual problems that may cause or have caused an adverse event for the patient or healthcare workers. Risk can be managed by using several approaches, including the FMEA and the FMECA [8,9], morbidity and mortality review, clinical auditing [10], significant event auditing, the London Protocol, the SHELL model, and root cause analysis [11]. The information collected after identifying and analyzing the biases present in the clinical care process is preparatory to the introduction of future risk prevention strategies, aimed at improving the quality of care. This objective is achieved through the introduction or implementation of procedures and protocols, by ensuring continuous training for healthcare workers and introducing new technologies [12].

Nowadays, the introduction and development of new technologies in the healthcare sector is going to transform medical service and offer the opportunity to promote harm minimization. Artificial intelligence (AI), the application of which has grown exponentially in various sectors in recent times, fits well in this context [13].

AI, coined in 1956 [14], is a modern informatics technology that simulates human behavior and makes devices efficient for achieving tasks that usually require skilled human intelligence. It is used in different fields and over the past few years, AI application in medicine has been growing [15]. AI encompasses several domains, such as machine learning (ML), which relies on different algorithms to learn and improve from experience without being programmed [16]; deep learning (DL), based on artificial neural networks [17]; or speech recognition, that is, the ability of a machine to convert a speech signal into a sequence of words, creating an interface between humans and technology [18]. AI can enhance medical achievements using automated clinical decision support (CDS), which assists health workers in making complex decisions in clinical practice by combining relevant patient information [19]. This application is facilitated by increasing diffusion of electronic medical record (EMR) and computerized provider order entry (CPOE) systems [20]. Moreover, natural language processing (NLP) is a technique able to convert unstructured text (e.g., medical records) into datasets easily analyzable by ML models [21]. Lastly, AI supports Internet of medical things (IoMT) technologies, which are bio-analytical tools capable of collecting, analyzing, and transmitting health data to increase the efficiency of human care [22].

Previous studies have reported that AI can improve the quality of healthcare [23], although its application still has numerous limitations [24].

This systematic review aims to investigate the state of the art regarding the impact of AI on clinical risk management processes. The International Classification for Patient Safety (ICPS), developed by the WHO, provides a taxonomy for the types of healthcare incidents that can occur, grouping them according to common characteristics, and facilitating benchmarks between results deriving from multiple sources, both at a national and an international level [25]. To simplify the analysis of the review outcomes and to motivate future standardized comparisons with any subsequent studies, the findings of the present review will be grouped according to the possibility of applying AI in the prevention of the different incident type groups as defined by the ICPS.

## 2. Materials and Methods

On 3 November 2023, a systematic review of the literature according to the Preferred Reporting Items for Systematic Reviews and Meta-Analyses (PRISMA) guidelines [26] was carried out using the SCOPUS and Medline (via PubMed) databases, using the following search strings: (artificial intelligence OR AI) AND (patient safety); (artificial intelligence OR AI) AND (risk management).

### 2.1. Inclusion and Exclusion Criteria

The inclusion criteria were as follows: case report; original article; short survey; article in English; human study; medical and nursing field; full text available; publication date between 1 January 2013 and 3 November 2023. A ten-year interval research was chosen to focus the review on the most recent studies.

The exclusion criteria were as follows: articles not in English; abstract; editorial; review; erratum; book chapter; note; conference paper. All articles focused on other topics were excluded.

### 2.2. Quality Assessment and Critical Appraisal

M.F. and G.B. independently evaluated the entire texts of the articles. The articles on which there was a disagreement were discussed with the senior investigator, R.L.R., for the final decision.

### 2.3. Risk of Bias

The main risk was linked to the keywords selected for the search strings. Therefore, the Kappa interobserver variability coefficient showed “almost perfect agreement” (0.80) [27].

### 2.4. Characteristics of Eligible Studies

A total of 297 articles were identified; 25 duplicate articles were removed, and 21 articles did not meet the inclusion criteria. After the selection process, 36 articles were included in the present systematic review.

## 3. Results and Discussion

Of the 297 articles found, 36 met the inclusion criteria (Figure 1).

The main features of each article included are summarized in Table 1.

### 3.1. Patient Safety Domains

In January 2009, the World Health Organization published a technical report providing the International Classification for Patient Safety (ICPS), a conceptual framework set out to allow for the codification of patient safety issues [62]. Specifically, “incident type” is one of the ten level classes comprising the framework, useful for categorizing patient safety incidents (clinical process, documentation, healthcare-associated infection, medication, blood, nutrition, oxygen, medical device, behavior, patient accidents, infrastructure, resources) [62].

Based on the aforementioned classification, results were grouped into ICPS patient safety domains to facilitate analysis from a risk management perspective. The studies included in this review allowed for the identification of three main domains: clinical process, healthcare-associated infection, and medication. Furthermore, the topic of incident reporting was discussed in an additional paragraph, since thirteen studies focus on this issue.

#### 3.1.1. Clinical Process

In recent times, AI has become increasingly prominent in the healthcare field, mainly in diagnosing, managing, treating, and screening pathologies [63,64,65,66]. Consequently, it is intuitive that the use of AI, in a clinical risk management perspective, concerns mainly these aspects of the Clinical process. By applying the ICPS classification, the use of AI in studies aimed at preventing the misinterpretation of radiographic investigations [28,29,56] or the operating field [30], translates into the potential advantage of avoiding inadequate treatments and procedures other than incorrect interventions.

Automated voice recognition software [32] and data analysis [33] have proven valuable in recognizing pathologies such as stroke and cancer, facilitating early detection and avoiding a delay in treatment. CDS systems [31,36] or intelligent checklists [59] can avoid harm to patients deriving from inadequate treatment or non-conformity to guidelines and good clinical practices.

#### 3.1.2. Healthcare-Associated Infections

Healthcare-associated infections (HCAIs) represent a major public health problem, with significant impacts on patients [67]. According to the Agency for Healthcare Research and Quality, HCAIs, and in particular sepsis, represent one of the ten leading causes of death in the United States [68]. Hence, as mortality in septic patients increases proportionally with the delay of antibiotic treatment, early prediction is crucial for timely interventions.

Machine learning algorithms applied to datasets can map several variables to predict the risk of Surgical Site Infections and sepsis [34,62] that could be undetected by healthcare workers, leading to a considerable reduction of morbidity and mortality-related infections. However, if the ML system is applied on a different dataset from the one on which it is deployed, the so-called dataset shift can occur, leading to the AI’s ineffective performance. For this reason, a clinician’s supervision is always necessary to evaluate any discrepancy between the clinical evaluation and the AI prediction or external validation to ascertain the general applicability of the ML system [61].

#### 3.1.3. Medication

An adverse drug event (ADE) consists of a medical therapy error that causes harm to the patient. It is estimated that this type of adverse event affects approximately 1 out of every 30 patients in health care [69].

Medication errors are due to different factors, including the human factor [70]. A very common error in clinical practice related to the human factor is prescription error [71]. One of the best-known human factor-related prescription error examples is that of look-alike or sound-alike (LASA) drugs, in which the error occurs due to orthographic or phonetic similarity or packaging between drugs [72,73]. AI could be useful in clinical practice to prevent so-called “look-alike” errors due to similar packaging between different medications by applying deep learning to drug images [52,58].

Furthermore, the error due to the simultaneous prescription of drugs that have the same effect can be avoided through the use of CDS systems which generate an alarm capable of alerting the healthcare professional [35]. However, these systems require high accuracy and a low number of alerts burns and false positives to avoid generating further stress in the healthcare worker, which would contribute to fueling the risk of errors related to the human factor [60].

Several studies have demonstrated that over half of ADEs occur during drug ordering [74]. To promote the best pharmaceutical care, it is essential to carry out a patient’s medication review, which is a critical evaluation of the drugs assumed to evaluate potential interactions and consequent adverse reactions. However, this part of clinical practice is often not exhaustive and takes a long time [75]. Machine learning combined with CDS systems or applied to data extracted from electronic health records are able to support pharmacovigilance activities and appears to be safe in performing medication reviews [37,54,55]. In addition, natural language processing linked to clinical notes can determine the reason for the pharmaceutical prescription to verify its appropriateness [57].

Computerized provider order entry (CPOE) systems reduce the risk of misinterpretation of the pharmacological order, as the integration of these systems with electronic health records promotes coordination of drug ordering, and cooperation between healthcare professionals allows for contextual collaboration between healthcare staff [74,75,76]. On the other hand, the interaction of the healthcare worker with the CPOE has proven to be a source of error [77,78], potentially hesitating in an ADE. In this regard, the prevention of these adverse events can be promoted through the use of machine learning systems capable of identifying the factors predisposing to medication ordering errors [46].

#### 3.1.4. Incident Reporting Systems

In healthcare facilities, incident reporting systems are essential for managing clinical risk by notifying providers of adverse events [79]. It consists of reporting adverse events, near misses, risks, and potentially unsafe conditions to healthcare professionals [80], including falls, HCAIs, transfusion [81], and patients’ and operators’ aggressions [82]. By using databases, healthcare facilities can identify, map, and analyze adverse events that occur to prevent them from occurring again. One of the limitations of this method concerns the inexperience of the different categories of healthcare professionals who carry out the reporting [83]. It has been noted in several studies that an absence of codified terminology is one of the main obstacles [84,85], resulting in an under-analysis of the event and, consequently, an inability to learn from it. Due to the high volume of data collected by the IT systems responsible for these purposes, using free text for reporting these events reduces its effectiveness due to the difficulty in aggregating the data [86]. Of the 36 articles included in this systematic review, 13 [38,39,40,41,42,43,44,47,48,49,50,51,53] investigate the applications of AI to improve the efficiency of incident reporting systems. The studies predominantly focus their attention on the possibility of standardizing events according to their type and severity. Furthermore, machine learning systems can evaluate the reporting rates of adverse events and estimate the risk of under-reporting [48] or to analyze contributing factors that predispose to their genesis [53].

The use of AI in this regard has a dual purpose. On the one hand, it reduces the workload of human work risk management staff, allowing them to dedicate themselves to other activities to implement patient safety [47,87]. On the other hand, the conversion of unstructured data into structured information has proven effective in identifying situations that are potentially fatal or capable of causing serious harm [51], prioritizing adverse events with significant consequences for patients [40]. The potential exclusive application of AI in real-world settings requires further studies, since a part of a reported event can be related to more than one type of accident [38]. In addition, the use of abbreviations or acronyms requires manual review [43,49]. At present, AI systems are not able to perfectly replace manual review [39], and the machine can only act as a support to risk management experts.

## 4. Conclusions

This review highlighted that AI can be applied transversely in various clinical contexts to enhance patient safety and facilitate the identification of errors. To facilitate the analysis of the present review outcome and to enable comparison with future systematic reviews, it was deemed useful to refer to a pre-existing taxonomy for the identification of adverse events.

For this purpose, it was considered convenient to use ICPS classification, which includes ten classes of “incident type”. The main fields of application, according to the codification mentioned above, concern the prevention of errors concerning clinical processes, medication errors and the development of HAIs.

Additionally, the results of the present study highlighted the usefulness of AI not only for risk prevention in clinical practice, but also in improving the use of an essential risk identification tool, which is incident reporting. It follows that ICPS classification could be limiting for the analysis of the application of AI to clinical risk management systems, as it relates to the clinical aspects of healthcare risk. For this reason, the taxonomy of the areas of application of AI to clinical risk processes should include an additional class relating to risk identification and analysis tools.

The advantages of using AI in risk management systems translate into a reduction in the workload of the risk manager, who can devote more time to developing procedures and paths to prevent mistakes, and the healthcare staff, who can spend more time with patients. However, aside from diminishing existing risks, as in the case of dataset shift, there is also the possibility of introducing new risks, such as false positive alerts that increase cognitive stress, thereby enabling human error. Furthermore, its use necessarily requires human supervision. To conclude, AI appears to be a promising tool to improve clinical risk management, although its use requires human supervision and cannot completely replace human skills.

## Figures and Tables

**Figure 1 healthcare-12-00549-f001:**
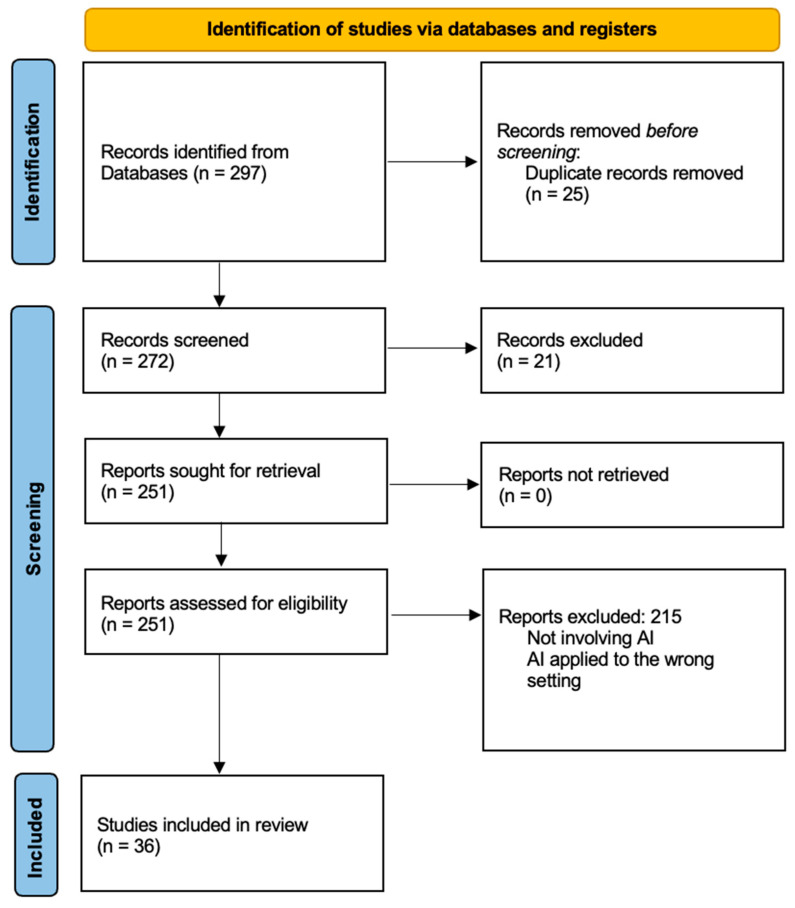
PRISMA flow diagram of this systematic review.

**Table 1 healthcare-12-00549-t001:** Main results of the systematic review.

Reference	Aim of the Study	Findings	Safety Domain
Drozdov et al. [28]	To develop a deep learning model for detection of nasogastric tube malposition on chest radiographs	The developed deep learning tool for detection of Naso Gastric Tube (NGT) misplacement on chest radiographs may lead to more rapid clinical image reviews, reducing interpretation time per image	Clinical process
Bowness et al. [29]	To evaluate the use of an assistive AI device to facilitate image acquisition for regional anesthesia	Use of an AI assistive ultrasound device was associated with improved ultrasound image acquisition and interpretation	Clinical process
Hameed et al. [30]	To evaluate the value of AI model which identifies safe and dangerous zones of dissection during laparoscopic cholecystectomy	AI is a useful tool for providing support and feedback to surgeons	Clinical process
Datar et al. [31]	To evaluate the decision-making impact of an artificial intelligence-enabled prognostic test in the management of diabetic kidney disease	The test has greater relative importance to primary care physicians (PCPs) than albuminuria and estimated glomerular filtration rate (eGFR) do in making treatment decisions	Clinical process
Scholz et al. [32]	To analyze the potential impact an automatic speech recognition (ASR) could have on stroke recognition at emergency medical services	An ASR can potentially improve stroke recognition by Emergency Medical Dispatchers (EMDs) and subsequent stroke treatment	Clinical process
Torrente et al. [33]	To present an AI-based solution tool for cancer patients’ data analysis and improve their management	The platform can provide real-time feedback by assessing risk of relapse, performing a stratification of patients, and predicting response to a certain treatment or utility of follow-up tests	Clinical process
Scala et al. [34]	To evaluate use of different artificial intelligence models to predict surgical site infections according to different risk factors	The K-nearest neighbors algorithm better handled the imbalanced dataset observing the highest accuracy value	Healthcare-associated infection
Brown et al. [35]	To use the electronic prescribing system to identify unintentional prescription of low molecular weight heparins (LMWHs) to patients prescribed direct-acting anticoagulants (DOACs).	The anticoagulant alert prevented duplicate anticoagulant prescription	Medication
Festor et al. [36]	To evaluate the safety of AI-based clinical decision support systems in sepsis treatment	AI consistently leads to a lower number of unsafe decisions in different clinical scenarios	Clinical process
Levivien et al. [37]	To attest to whether prescriptions with low risk of drug-related problems (DRPs) ruled out by a digital tool using machine learning with AI in everyday practice were effectively free of any DRPs with potentially severe clinical impact	This hybrid decision support tool was shown to be accurate in detecting DRPs in daily practice	Medication
Wang et al. (1) [38]	To evaluate text classification using binary classifier ensembles to automate the identification of patient safety incident reports by type and severity	Binary classifier ensembles appear to be a feasible method for identifying incidents by type and severity level. Automated identification should enable safety problems to be detected and addressed in a more timely manner	Incident reporting systems
Evans et al. [39]	To test the capability of autonomous classifying of free text within patient safety incident reports to determine incident type and the severity of harm outcome	Supervised machine learning can be used to classify patient safety incident report categories	Incident reporting systems
Wang et al. (2) [40]	To evaluate the feasibility of a convolutional neural network (CNN) with word embedding to identify the type and severity of patient safety incident reports	CNN has potential to be applied in a real-world setting as a first step for group incidents when human resources are lacking	Incident reporting systems
Ozonoff et al. [41]	To describe and evaluate an approach to surveillance of safety events captured in electronic data sources, including structured data fields within the EMR, and unstructured data, including clinical notes	When provided a high-quality training set, the Support Vecture Tracking (SVM) model could classify unstructured free-text notes with reasonably high sensitivity and specificity. This is important to reduce the necessary human review that follows classification	Incident reporting systems
Wang et al. (3) [42]	To evaluate the feasibility of using Unified Medical Language System (UMLS) semantic features for automated identification of reports about patient safety incidents by type and severity	UMLS-based semantic classifiers were effective in identifying incidents by type and extreme-risk events	Incident reporting systems
Fong et al. [43]	To identify health information technology (HIT)-related events from patient safety event (PSE) report free-text descriptions	The feature constraint model provides a method to identify HIT-related patient safety hazards using a method that is applicable across healthcare systems with variability in their PSE report structures	Incident reporting systems
Lu et al. [44]	To evaluate various text classification methods of adverse nursing events based on deep learning	The results show the exceptional performance of the proposed mechanism in terms of various evaluation metrics	Incident reporting systems
King et al. [45]	To use machine learning (ML) models to predict erroneous medication orders and identify their contributing factors	The methodological approach of using ML algorithms for predicting medication errors has two potential applications: to identify factors associated with order entry errors that potentially represent generalizable knowledge for mitigating such errors and to guide patient safety efforts that are targeted towards medication orders within high-risk contexts	Medication
Fong et al. (2). [46]	To apply active learning techniques to support classification of patient safety event reports as HIT-related	Active learning can be used to identify HIT-related events from large datasets where human annotation is a major barrier to understanding trends and patterns in the data	Incident reporting systems
Barmaz et al. [47]	To propose a method to compute the probability of AE underreporting that could complement a machine learning mode developed to enhance patients’ safety while reducing the need for on-site and manual clinical quality assurance (QA) activities in clinical trials	This approach reduces the need for on-site audits, shifting focus from source data verification to pre-identified, higher risk areas; it will enhance further QA activities for safety reporting from clinical trials and generate quality evidence during pre-approval inspections	Incident reporting systems
Zhou et al. [48]	To propose an automated pipeline to identify medication event reports and reduce the workload of patient safety experts	The pipeline is expected to save time and reduce the workload for clinicians to analyze event reports and better discover valuable information from the reports to facilitate the development of strategies for preventing medication events	Incident reporting systems
Wong et al. [49]	To develop a medication rights detection system using natural language processing and deep neural networks to automate medication incident identification using free-text incident reports	This study developed a medication rights detection system via DNN to automate medication incident identification using free-text incident reports and provide reference guidelines for training DNN models to classify patient safety incidents The deep learning method shows promise for the efficient exploration of textual reports of medication incidents	Incident reporting systems
Yang et al. [50]	To develop a deep learning model to identify allergic reactions in the free-text narrative of hospital safety reports and evaluate its generalizability, efficiency, productivity, and interpretability	A deep learning model can accurately and efficiently identify allergic reactions using free-text narratives written by a variety of health care professionals; this model could be used to improve allergy care, potentially enabling real-time event surveillance and guidance for medical errors and system improvement	Incident reporting systems
Ting et al. [51]	To understand how identification confusion of look-alike images by human occurs through the cognitive counterpart of deep learning solutions and to suggest further solutions to approach them	This model outperformed identification using conventional computer vision solutions and could assist pharmacists in identifying drugs while preventing medication errors caused by look-alike blister packages	Medication
Tabaie et al. [52]	To explore the use of natural language processing (NLP) algorithms to categorize contributing factors from patient safety events (PSEs)	Applying the information-rich sentence selection algorithm boosted contributing factor categorization performance	Incident reporting systems
Zhao et al. [53]	To explore ways of using structured electronic health record data that can be exploited to detect a wide range of adverse drug events (ADEs), which could be adopted in a general decision support system that alerts for potential ADEs	Machine learning can be applied to electronic health records for the purpose of detecting adverse drug events and proposed solutions	Medication
Corny et al. [54]	To test the accuracy of a hybrid clinical decision support system in prioritizing prescription checks to improve patient safety and clinical outcomes by reducing the risk of prescribing errors	This novel hybrid decision support system improved the accuracy and reliability of prescription checks in a hospital setting	Medication
Lee et al. [55]	To propose a fully automated deep learning system with a cascading segmentation AI system containing two fully convolutional neural networks for detecting a peripherally inserted central catheter (PICC) line and its tip location	This system could help speed confirmation of PICC position and further be generalized to include other types of vascular access and therapeutic support devices	Clinical process
Li et al. [56]	To acquire a comprehensive structured and coded knowledge base of indications for medications, and to develop methods that determine the reasons for medication uses in the Electronic Health Record (EHR) using the knowledge base	This pilot study demonstrated that linking external drug indication knowledge to the EHR for determining the reasons for medication use was promising	Medication
You et al. [57]	To propose a novel two-stage induced deep learning (TSIDL) method to classify diverse packaging of drugs efficiently and accurately	The proposed TSIDL method significantly improves the performance of classifying visually similar drugs with diverse packaging types	Medication
De Bie et al. [58]	To compare paper checklists (control) with a dynamic (digital) clinical checklists	A digital checklist improved compliance with best clinical practice compared with a paper checklist during ward rounds in a mixed ICU	Clinical process
Segal et al. [59]	To evaluate the accuracy, validity, and clinical usefulness of medication error alerts generated by a novel system using outlier detection screening algorithms	A clinical decision support system that used a probabilistic, machine learning approach based on statistically derived outliers to detect medication errors generated clinically useful alerts The system had high accuracy, low alert burden and low false-positive rate, and led to changes in subsequent orders	Medication
Zhu et al. [60]	To understand the generalizability of a machine learning automated surgical site infection (SSI) detection algorithms	SSI detection machine learning algorithms developed at 1 site were generalizable to another institution	Healthcare-associated infection
Lind et al. [61]	To develop a full risk factor (demographic, transplant, clinical, and laboratory factors) and clinical factor-specific automated bacterial sepsis decision support tool for recipients of allogeneic hematopoietic cell transplants with potential bloodstream infections	Compared with existing tools and a clinical factor-specific tool, the full decision support tool had superior prognostic accuracy for the primary (high sepsis risk bacteremia) and secondary (short-term mortality) outcomes in inpatient and outpatient settings	Healthcare-associated infection

## Data Availability

Data is contained within the article.

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
