# Peer review of "Risk Management and Patient Safety in the Artificial Intelligence Era: A Systematic Review"

_healthcare, 2024, doi:10.3390/healthcare12050549_

Round 1

Reviewer 1 Report

Comments and Suggestions for Authors

Dear Authors,

I hope you are doing well. I am writing to you to give you feedback on the manuscript submitted for consideration in the healthcare journal.

After a thorough review of the paper, I would like to offer several constructive suggestions to enhance the quality and comprehensiveness of the research:

1.    Search Term Expansion: While the search strategy employed in the study aimed to capture relevant articles on artificial intelligence (AI), patient safety, and risk management, there is a potential limitation in the chosen search terms. The search strings may inadvertently exclude studies that utilize different terminology or keywords to address similar topics. I recommend expanding the search terms to encompass synonyms, related terms, and variations of the key concepts. For example, alongside "artificial intelligence" and "AI," terms like "machine learning," "deep learning," and "neural networks" should be included. Additionally, terms such as "healthcare," "clinical," "hospital," or "medical" can be integrated to focus the search on literature related to healthcare settings.

Example: ("artificial intelligence" OR "AI" OR "machine learning" OR "deep learning" OR "neural networks" OR "computational intelligence" OR "natural language processing" OR "expert systems") AND ("patient safety" OR "healthcare quality" OR "medical errors" OR "adverse events" OR "patient outcomes" OR "clinical risk" OR "hospital safety") AND ("risk management" OR "risk assessment" OR "risk mitigation" OR "safety culture" OR "patient harm prevention" OR "adverse event prevention" OR "error reduction" OR "incident reporting")

2.    Justification for Publication Timeframe: The manuscript should clarify the rationale behind setting the publication timeframe between January 1, 2013, and November 3, 2023. Was the search volume significantly low before 2013, or are there other reasons for this specific timeframe selection?

3.    Limitations of ICPS Framework: The paper should acknowledge that the International Classification for Patient Safety (ICPS) framework may cover only some aspects of patient safety incidents, as it includes only ten level classes. This limited scope may only partially capture the complexity and diversity of patient safety issues in healthcare settings.

4.    Inclusion of Quantitative Data: The manuscript should incorporate quantitative data where applicable. Adding statistical information and discussing its implications can enhance the depth of analysis and provide more robust insights into the research findings.

5.    Clarification of Terminology: The manuscript should clarify the meaning of "IA" in the screening section of the PRISMA chart. Does it refer to "AI"?

Addressing these points will significantly strengthen the manuscript and contribute to its scholarly value. Thank you for considering these suggestions, and I look forward to the revised version of the manuscript.

Best Regards.

Author Response

Reviewer 1:

1.Search Term Expansion: While the search strategy employed in the study aimed to capture relevant articles on artificial intelligence (AI), patient safety, and risk management, there is a potential limitation in the chosen search terms. The search strings may inadvertently exclude studies that utilize different terminology or keywords to address similar topics. I recommend expanding the search terms to encompass synonyms, related terms, and variations of the key concepts. For example, alongside "artificial intelligence" and "AI," terms like "machine learning," "deep learning," and "neural networks" should be included. Additionally, terms such as "healthcare," "clinical," "hospital," or "medical" can be integrated to focus the search on literature related to healthcare settings.

Example: ("artificial intelligence" OR "AI" OR "machine learning" OR "deep learning" OR "neural networks" OR "computational intelligence" OR "natural language processing" OR "expert systems") AND ("patient safety" OR "healthcare quality" OR "medical errors" OR "adverse events" OR "patient outcomes" OR "clinical risk" OR "hospital safety") AND ("risk management" OR "risk assessment" OR "risk mitigation" OR "safety culture" OR "patient harm prevention" OR "adverse event prevention" OR "error reduction" OR "incident reporting")

We thank the reviewer for their suggestion. We believe that adding additional terms to the search string, such as "machine learning" or "neural networks," is unnecessary because scientific articles containing these keywords already include the keyword "AI."

 2.Justification for Publication Timeframe: The manuscript should clarify the rationale behind setting the publication timeframe between January 1, 2013, and November 3, 2023. Was the search volume significantly low before 2013, or are there other reasons for this specific timeframe selection?

Thank you for your comment. We made the required integrations.

3.Limitations of ICPS Framework: The paper should acknowledge that the International Classification for Patient Safety (ICPS) framework may cover only some aspects of patient safety incidents, as it includes only ten level classes. This limited scope may only partially capture the complexity and diversity of patient safety issues in healthcare settings.

Thanks for the suggestion. We really appreciated your help in improving our work.

4. Inclusion of Quantitative Data:The manuscript should incorporate quantitative data where applicable. Adding statistical information and discussing its implications can enhance the depth of analysis and provide more robust insights into the research findings.

Thank you for your comment.

  1.  Clarification of Terminology:The manuscript should clarify the meaning of "IA" in the screening section of the PRISMA chart. Does it refer to "AI"?

Thanks for highlighting this. We corrected the typo.

Addressing these points will significantly strengthen the manuscript and contribute to its scholarly value. Thank you for considering these suggestions, and I look forward to the revised version of the manuscript.

Reviewer 2 Report

Comments and Suggestions for Authors

Authors need to address the following suggestions. 

1. Discussion section need to be strengthened by clearly highlighting the research gaps observed in the reported literature. 

2. Manuscript has to be proofread for typos. Ex. refer section 3.1

3. Section 4 is missing. 

4. Group citations need to be avoided.

5. Most of the references cited were old. Authors need to include the discussions from recent research works. 

Comments on the Quality of English Language

Grammar check and typos need to be corrected. 

Author Response

Reviewer 2:

1. Discussion section need to be strengthened by clearly highlighting the research gaps observed in the reported literature. 

Thank you for your comment. We made the required integrations.

2. Manuscript has to be proofread for typos. Ex. refer section 3.1

We thank the reviewer for their comment. The entire manuscript has undergone proofreading.

3. Section 4 is missing. 

Thanks for highlighting this. We corrected the typo.

4. Group citations need to be avoided.

We thank the reviewer for their comment. Group citations are accepted within a limit of 20% of the paper. In this article, there are 4 group citations out of a total of 88 citations, which equals 4.5%, a value considered acceptable.

5. Most of the references cited were old. Authors need to include the discussions from recent research works. 

We thank the reviewer for their comment. We have included updated citations.

Reviewer 3 Report

Comments and Suggestions for Authors

Dear Authors, thank you for submitting this exciting manuscript. I recommend several points that need to be considered in your article. I hope it might help you with the improvement.

I enjoyed the attention to using artificial intelligence in clinical risk management but understanding the relevance and existence of the knowledge gap still needs to be more explicit. The introduction section needs to be more accurate; some critical issues deserve a more significant deepening, starting with a more detailed general description of clinical risk management and the relative tools. The gap and the study's aims need to be adequately explained.

Please insert references in lines 51, 55, and 69 at the end of the paragraph.

In the materials and methods section, the search string has been limited to the indicated combinations, and no other possibilities have been included, such as "clinical risk management" or "machine learning." Please explain the choice of the string, why the previous systematic reviews and the chapters of the books were excluded, and how the data were collected. 

Please insert the reference in line 111 at the end of the paragraph.

The current system for classifying incidents needs to align with the review outcomes. It is suggested that a revised classification system that includes incident reporting aspects be introduced since incident reporting accounts for more than one-third of the review results. Please clarify the reasons for the current classification system or propose an alternative one better aligned with the review results.

The data reported in lines 145-147 may not be accurate or up-to-date. Please see WHO's latest update.

Please insert the reference in line 165 at the end of the paragraph.

A review of the introduction of paragraph 3.1.4 on incident reporting needs to include references (e.g., lines 179-185). In addition, the aspects that the incident reporting system can cover should be discussed in more detail (please note DOI: 10.3390/ijerph17176267 and DOI: 10.1097/NMD.0000000000001504)

Please review the conclusions and abstract in light of the guidance provided. In particular, the abstract needs more clarification, especially regarding the study's aims, the results, and the discussion. Please enrich lines 26-30 and explain the study's aims better.

Moreover, several things could be improved in the references section, such as citation No. 6 and citation No. 10, which are the same.

Please also give reasons for consistency in the text of reference No. 70.

Comments on the Quality of English Language

Minor editing of the English language is required.

Author Response

Reviewer 3:

I enjoyed the attention to using artificial intelligence in clinical risk management but understanding the relevance and existence of the knowledge gap still needs to be more explicit. The introduction section needs to be more accurate; some critical issues deserve a more significant deepening, starting with a more detailed general description of clinical risk management and the relative tools. The gap and the study's aims need to be adequately explained.

Thanks for highlighting this. We have modified the introduction.

Please insert references in lines 51, 55, and 69 at the end of the paragraph.

Thank you for the comments.

In the materials and methods section, the search string has been limited to the indicated combinations, and no other possibilities have been included, such as "clinical risk management" or "machine learning." Please explain the choice of the string, why the previous systematic reviews and the chapters of the books were excluded, and how the data were collected. 

Previous systematic reviews and book chapters were excluded as a systematic review should include exclusively original research articles.

Please insert the reference in line 111 at the end of the paragraph.

The current system for classifying incidents needs to align with the review outcomes. It is suggested that a revised classification system that includes incident reporting aspects be introduced since incident reporting accounts for more than one-third of the review results. Please clarify the reasons for the current classification system or propose an alternative one better aligned with the review results.

The data reported in lines 145-147 may not be accurate or up-to-date. Please see WHO's latest update.

Please insert the reference in line 165 at the end of the paragraph.

A review of the introduction of paragraph 3.1.4 on incident reporting needs to include references (e.g., lines 179-185). In addition, the aspects that the incident reporting system can cover should be discussed in more detail (please note DOI: 10.3390/ijerph17176267 and DOI: 10.1097/NMD.0000000000001504)

Thank you for your comment. We modified as requested.

82. Ferorelli, D.; Solarino, B.; Trotta, S.; Mandarelli, G.; Tattoli, L.; Stefanizzi, P.; Bianchi, F.P.; Tafuri, S.; Zotti, F.; Dell’Erba, A. Incident Reporting System in an Italian University Hospital: A New Tool for Improving Patient Safety. J. Environ. Res. Public Health 2020, 17, doi:10.3390/ijerph17176267.

83. Mele, F.; Buongiorno, L.; Montalbò, D.; Ferorelli, D.; Solarino, B.; Zotti, F.; Carabellese, F.F.; Catanesi, R.; Bertolino, A.; Dell’Erba, A.; et al. Reporting Incidents in the Psychiatric Intensive Care Unit: A Retrospective Study in an Italian University Hospital. Nerv. Ment. Dis. 2022, 210, 622–628, doi:10.1097/NMD.0000000000001504.

Please review the conclusions and abstract in light of the guidance provided. In particular, the abstract needs more clarification, especially regarding the study's aims, the results, and the discussion. Please enrich lines 26-30 and explain the study's aims better.

Moreover, several things could be improved in the references section, such as citation No. 6 and citation No. 10, which are the same.

It should be noted that citations are different, as they refer to two distinct studies regarding the application of FMEA method.

Please also give reasons for consistency in the text of reference No. 70.

We thank the reviewer for the comments.

Round 2

Reviewer 1 Report

Comments and Suggestions for Authors

Thanks for addressing the comments.

Reviewer 2 Report

Comments and Suggestions for Authors

Authors have addressed my suggestions. 

Comments on the Quality of English Language

Minor changes

Reviewer 3 Report

Comments and Suggestions for Authors

Dear Authors,

I have completed the second review of your manuscript, and I want to congratulate you on your exceptional work in revising it. You have responded comprehensively and accurately to all concerns raised during the first review, demonstrating remarkable dedication to refining your research. Your revisions have significantly improved the text's clarity and the presentation of results. Thank you for your commitment and collaboration.

Best regards.